# Antibiotic Use and Treatment Outcomes among Children with Community-Acquired Pneumonia Admitted to a Tertiary Care Public Hospital in Nepal

**DOI:** 10.3390/tropicalmed6020055

**Published:** 2021-04-20

**Authors:** Bhishma Pokhrel, Tapendra Koirala, Dipendra Gautam, Ajay Kumar, Bienvenu Salim Camara, Saw Saw, Sunil Kumar Daha, Sunaina Gurung, Animesh Khulal, Sonu Kumar Yadav, Pinky Baral, Meeru Gurung, Shrijana Shrestha

**Affiliations:** 1Patan Hospital, Patan Academy of Health Sciences, Lalitpur 44700, Nepal; sunilkumardaha@pahs.edu.np (S.K.D.); sunaina.the.g@gmail.com (S.G.); anmshkbassnet@gmail.com (A.K.); sonukumaryadav@pahs.edu.np (S.K.Y.); drmeeru@hotmail.com (M.G.); shrijanashrestha@pahs.edu.np (S.S.); 2Department of Health Services, Ministry of Health and Population, Kathmandu 44600, Nepal; 3World Health Emergencies Program, WHO Country Office, Kathmandu 44600, Nepal; gtmdipen@gmail.com; 4International Union against Tuberculosis and Lung Disease, South-East Asia Office, New Delhi 110016, India; AKumar@theunion.org; 5International Union against Tuberculosis and Lung Disease, 75006 Paris, France; 6Yenepoya Medical College, Yenepoya, Mangaluru 575018, India; 7Central National de Formation et de Recherche en Santé Rurale de Maferinyah, Forécariah 4090, Guinea; bscamara@maferinyah.org; 8Department of Medical Research, Ministry of Health and Sports, Yangon 05081, Myanmar; sawsaw@mohs.gov.mm; 9Modern Technical College, Sanepa, Lalitpur 44700, Nepal; baralpinky@gmail.com

**Keywords:** community-acquired pneumonia, CAP, antibiotic use, treatment outcome, operational research, SORT IT

## Abstract

In the era of growing antimicrobial resistance, there is a concern about the effectiveness of first-line antibiotics such as ampicillin in children hospitalized with community-acquired pneumonia. In this study, we describe antibiotic use and treatment outcomes among under-five children with community-acquired pneumonia admitted to a tertiary care public hospital in Nepal from 2017 to 2019. In this cross-sectional study involving secondary analysis of hospital data, there were 659 patients and 30% of them had a history of prehospital antibiotic use. Irrespective of prehospital antibiotic use, ampicillin monotherapy (70%) was the most common first-line treatment provided during hospitalization followed by ceftriaxone monotherapy (12%). The remaining children (18%) were treated with various other antibiotics alone or in combination as first-line treatment. Broad-spectrum antibiotics such as linezolid, vancomycin, and meropenem were used in less than 1% of patients. Overall, 66 (10%) children were required to switch to second-line treatment and only 7 (1%) children were required to switch to third-line treatment. Almost all (99%) children recovered without any sequelae. This study highlights the effectiveness of ampicillin monotherapy in the treatment of community-acquired pneumonia in hospitalized children in a non-intensive care unit setting.

## 1. Introduction

Pneumonia remains the most important cause of child mortality beyond the neonatal period globally [1], and also the most important cause of under-five mortality in south-east Asia [2]. It was estimated that around 65,000 children died of pneumonia in 2016 globally with more than 85% occurring in south-east Asian countries. The annual incidence of pneumonia among under-five children in south-east Asia is estimated to be 0.36 episodes per child [3]. In Nepal, pneumonia accounted for 10% of all childhood acute respiratory infections (ARI) in 2017 with an annual incidence of 66 cases per 1000 under-five children [4]. Previous studies from tertiary care settings in Nepal show that most of the pneumonia in under-five children were viral in origin and those with bacterial etiology were sensitive to the penicillin group of antibiotics [5,6].

A clinical case management approach formulated by the World Health Organization (WHO) is in place to reduce pneumonia-related deaths and improve access to life-saving antibiotics for children with community-acquired pneumonia (CAP), and to prevent unnecessary antibiotic use and hospitalizations in those who can be safely managed as outpatients [7]. WHO recommends facility-based parenteral ampicillin and gentamicin as the empirical therapy to treat all severe pneumonia whereas home-based oral amoxicillin for all non-severe pneumonia on an outpatient basis is recommended. However, large variation is seen in antibiotic use within and across countries, with inappropriate use of antibiotics common in low- and middle-income countries. Increased utilization of antibiotics without prescription has been reported in developing countries, with 44% to 97% of patients receiving inappropriate antibiotics [8,9,10]. The irrational use and overuse of antibiotics may lead to the emergence of antimicrobial resistance (AMR), especially in resource-limited countries, due to unregulated distribution of antibiotics in the population, inappropriate health provider practices such as the wrong prescription of antibiotics, self-medication, and incomplete intake of prescribed antibiotic doses by the patients [11].

With the growing threat of AMR in south-east Asia [12], understanding the pattern of antibiotics used to treat common pediatric illnesses like CAP is crucial. There is limited information on antibiotic regimens used to treat CAP in tertiary level hospitals in Nepal. Hence, we conducted an operational research study, among the most vulnerable age group (2–59 months) admitted with CAP to the Patan hospital in Nepal, with the following objectives: i) to describe the socio-demographic and clinical characteristics, ii) to determine the yield of blood culture, and iii) to describe the patterns of antibiotic regimens used for treatment during hospitalization and treatment outcomes.

## 2. Materials and Methods

### 2.1. Study Design

This was a cross-sectional study using routinely collected secondary data from the electronic records and paper-based patient files.

### 2.2. Study Setting

#### 2.2.1. General Setting

Nepal is a small landlocked country in south Asia, lying between two large countries, India and China. It is composed of seven federal provinces with a population of 30 million people (as of 2017). About 35% of the population are children aged less than 15 years [13] and about 40% live below the poverty line [14]. The health system of Nepal consists of 125 public hospitals and 1822 non-public hospitals. There are three levels of health facilities ranging from local (primary), to provincial (secondary), to central (tertiary). From 2013 to 2018, the per capita government spending on health gradually increased from Nepalese rupees (NPR) 966 to NPR 1819 (USD 9.8 to 17.7) [15].

#### 2.2.2. Specific Setting

The study was conducted at the Patan Hospital, a 640-bedded autonomous, not-for-profit, tertiary level public hospital located in the southern part of Lalitpur district of province 3. It is one of the WHO sentinel sites for the surveillance of invasive bacterial diseases, namely pneumonia, sepsis, and meningitis. The hospital has an average of 320,000 outpatient visits and 20,000 admissions annually.

#### 2.2.3. Protocol Used at the Hospital

Patan Hospital has a 54-bedded general pediatric ward apart from 12-bedded intensive care units (pediatric ICU and neonatal ICU) and 20-bedded neonatal nurseries. Identification of CAP is based on a clinical diagnosis by the pediatrician in any child with a history of cough and/or shortness of breath, with or without fever, and relevant findings on systemic examination aided by radiological evidence as defined by the Infectious Diseases Society of America (IDSA) [14]. For the treatment of CAP, various factors like the severity of illness, prior antibiotic use, underlying comorbidities, nutritional status, and radiological findings on chest radiograph play an important role in helping the pediatrician to decide which antibiotic to be used. The antibiotic used in the first 48 h is considered the first-line antibiotic. Ampicillin with or without amikacin, cloxacillin, and azithromycin are commonly used as the first-line antibiotics. If there is no clinical improvement by 48 h after starting first-line treatment or if there is rapid deterioration within 48 h, the antibiotic is changed (second-line treatment). Antibiotics like ceftriaxone, cefotaxime, chloramphenicol, vancomycin, and linezolid are often used as second-line antibiotics. If clinical improvement does not occur within 48 h of starting second-line antibiotics, the treating pediatrician considers upgrading the antibiotics to the third-line antibiotics. Meropenem and colistin are considered the third-line antibiotics. However, antibiotics that are considered as the second- or third-line may be used as first-line or second-line, especially when the child has a history of prehospital antibiotic use and clinically severe. Before starting antibiotic treatment, a blood sample is collected from each patient for culture and drug susceptibility test as a part of WHO invasive bacterial disease surveillance, which takes at least 72 h before results are available. If any bacterial pathogen is identified from the blood culture, the antibiotic is modified based on the culture and sensitivity pattern.

### 2.3. Study Population

We included all children aged 2–59 months with a diagnosis of CAP admitted to the pediatric ward of the Patan Hospital from 1 January 2017 to 31 December 2019. For this study, we defined CAP as "an acute infection of the pulmonary parenchyma that is associated with some symptoms of acute infection, accompanied by the presence of an acute infiltrate on a chest radiograph or auscultatory findings consistent with pneumonia in a patient not hospitalized or residing in a long-term care facility for more than 14 days before the onset of symptoms" as per IDSA guidelines [16]. The manifestations of acute infection include fever, cough, age-specific tachypnea, and lower chest wall indrawing; the auscultatory findings include abnormal breath sounds, wheezes, or crackles; and the radiographic evidence includes consolidation, other infiltrate, or pleural effusion. 

The exclusion criteria included:Children who had complicated pneumonia (as defined by the presence of significant effusion, empyema, necrotizing pneumonia, pneumothorax, severe or impending respiratory failure, and/or signs and symptoms of sepsis or shock) at the time of admission.Persistent (chronic) pneumonia syndromes (with symptoms for > 2 weeks).History suggestive of aspiration pneumonia or recurrent pneumonia.Pneumonia associated with chronic medical problems such as immunodeficiency such as diabetes mellitus, chronic kidney disease, bronchial asthma, lung malignancy, known HIV infection.Invasive mechanical ventilation within 14 days before the current hospital admission.Children who have taken antibiotics for a respiratory infection for more than 7 days before the hospital admission.History of cystic fibrosis, post-obstructive pneumonia, or active tuberculosis.

### 2.4. Data Variables and Sources

Data were collected between March and December 2020. Data variables included age, sex, date of hospital admission, chest radiograph findings, nutritional status, the severity of pneumonia (defined by the presence of any of the danger signs such as convulsions, lethargy or altered sensorium, persistent vomiting, poor oral intake, and cyanosis as per the WHO definition) [7], history of prehospital antibiotic use, blood culture results, antibiotics used in first-line, second-line, and third-line treatment, and treatment outcomes. Data sources included the patient files (both electronic and paper-based) and the laboratory register of the microbiology department.

### 2.5. Analysis and Statistics

The data collected from the patient charts were double entered and validated using EpiData version 3.1 (EpiData Association, Odense, Denmark). This was merged with the electronic data using a case identification number and a master dataset prepared for analysis. Data were reviewed for inconsistencies and cleaned before analysis. Data were analyzed using EpiData Analysis software (version 2.2.2.187). Descriptive analysis was conducted and presented as frequencies and proportions for categorical data.

## 3. Results

### 3.1. Sociodemographic and Clinical Characteristics

Overall, 659 children were admitted with CAP; 57% were females and 46% were infants aged less than 12 months. Of these, only 26% met the WHO criteria for severe pneumonia whereas 39% did not meet the WHO criteria for pneumonia. The majority of children (66%) had normal chest radiography findings. Among children with WHO-defined severe CAP, only 10% had positive findings in their chest radiography. Prehospital antibiotic use was seen in 30% of children (Table 1). Wheezing was seen in 214 (32%) children.

Most of the children reported a history of immunization using pneumococcal conjugate vaccine (PCV) (91%) and *Hemophilus Influenzae* B (Hib) (98%) vaccines, but the information on the number of doses of vaccine received was missing in 12% of children for PCV and 48% of children for Hib vaccine—this precluded assessment of completeness of vaccination appropriate to the age. Overall, the completeness of vaccination was better with PCV compared to the Hib vaccine (68% versus 48%).

### 3.2. Blood Culture Results

Of the 659 patients, 646 (98%) had their blood culture done and only 5 (0.8%) patients were culture-positive for bacterial pathogens (three patients with coagulase-negative *Staphylococcus* (CoNS), one *Streptococcus pneumoniae*, and one *Streptococcus viridans*). CoNS were considered “contaminants” and thus no treatment changes were required. Both *Streptococcus pneumoniae* and *Streptococcus viridans* were fully sensitive to the penicillin group of drugs including ampicillin.

### 3.3. Antibiotic Use during Hospitalization

The antibiotic regimens used as first-line treatment after hospital admission is depicted in Table 2. Irrespective of prehospital antibiotic use, ampicillin monotherapy (69.5%) followed by ceftriaxone monotherapy (12.3%) were the two most common first-line antibiotics regimens used. The remaining children (18%) were treated with other antibiotics as first-line treatment and there was a total of 23 different regimens used. Overall, 66 (10%) children were required to switch to second-line antibiotics and only 7 (1%) children were required to switch to third-line antibiotics. The proportion of children who received ampicillin monotherapy and subsequently required to upgrade to second-line and third-line were 11.6% and 0.4%, respectively, while that for the ceftriaxone monotherapy group was 4.9% and none, respectively (Table 3). Among children with WHO-defined severe pneumonia receiving ampicillin monotherapy, only 4.6% required second-line treatment. All children (except two who received oral azithromycin) were started on intravenous antibiotics. The overall use of antibiotics is shown in Table 4 which shows that Ampicillin was used in 80% of the children (either singly or in combination) followed by Ceftriaxone and Cloxacillin.

### 3.4. Treatment Outcomes

Of 659 children, 651 (99%) recovered. A total of five children left the hospital against medical advice while on treatment and two were transferred to the intensive care unit. One patient was sent home after 72 h of hospitalization on oral therapy as per the parent’s request.

## 4. Discussion

Antimicrobial resistance is a serious public health threat globally. The present study is one of the few from Nepal to carry out a situation analysis on the use of antibiotics to treat CAP in hospitalized children aged 2–59 months. In our study, the majority (70%) of the children were treated with ampicillin monotherapy and among them, nearly nine in ten children had an uneventful recovery without the need for antibiotic upgrade irrespective of prehospital antibiotic use. The use of ceftriaxone as a first-line antibiotic was low (12.3%). There exists significant variation in the use of empirical first-line antibiotics recommended by various guidelines in the treatment of CAP in hospitalized children. WHO’s recent recommendations emphasize the parenteral use of ampicillin and gentamicin in the treatment of severe CAP [7], whereas the American Academy of Pediatrics (AAP) recommends ampicillin as the first-line antibiotic for uncomplicated pediatric CAP requiring hospitalization [17].

In our study, nearly seven in ten children who met the WHO criteria for severe pneumonia were also treated with ampicillin monotherapy without any need for upgrade into second- or third-line antibiotics. Only 4.6% of those children needed an antibiotic upgrade. This finding is in agreement with the recommendation by the AAP for the treatment of uncomplicated pediatric CAP. We did not find any published literature comparing the treatment outcome of ampicillin alone versus ampicillin in combination with gentamicin in the treatment of pneumonia as per WHO classification. However, ampicillin has been compared with various other antibiotics in the treatment of uncomplicated CAP in hospitalized children. One study from Israel found comparable treatment outcomes between the penicillin/ampicillin group and cefuroxime group [18].

There have been several other studies that have shown the use of third-generation cephalosporins as the first-line treatment in pediatric CAP. Such broad-spectrum antibiotics have been shown to contribute to antimicrobial resistance and *Clostridium difficile* infection [17], and should be used judiciously only where there is a strong suspicion of highly penicillin-resistant *Streptococcus pneumoniae* or beta-lactamase producing *Haemophilus influenzae* [19]. Antibiotic use before the hospital admission was seen in 30% of the children in our study. Other studies from Asian countries like the Philippines (17–53%), Bhutan (30.4%), and Vietnam (49.1%) reported similar findings [20,21,22]. Easy availability and sale of antibiotics over-the-counter could be one of the main reasons for such a high prevalence of prehospital antibiotic use among the study participants, which might contribute to antimicrobial resistance in the future.

The yield of blood culture was extremely low (0.75%) in this study. Only two pathogens (*Streptococcus pneumoniae* and *Streptococcus viridians*) were isolated and they were fully susceptible to the penicillin group of drugs. Various studies have reported a wide range of bacteremia in pediatric CAP (0.8–17.4%) [23]. Therefore, studies conducted in non-ICU settings have recommended that routine use of blood culture may not be needed for most children hospitalized with CAP [24], as it has been found that viruses are the most common causes of CAP in healthy, immunized children, and bacteria account for ~15% of the cases of which *Streptococcus pneumoniae* is the most common pathogen [25,26]. The routine immunization against *Streptococcus pneumoniae* and *Haemophilus influenzae* type B in Nepal has resulted in a decrease in pediatric CAP secondary to these invasive bacterial infections, from 7.7% to 4% [23,27]. However, children admitted in ICU settings with complicated pneumonia may benefit from the routine performance of blood cultures [28].

Though all the patients enrolled in our study had their chest radiography performed, only one-third of them had radiological evidence of pneumonia (32.3%). Among the children with WHO severe pneumonia, the chest radiograph was normal in 61.8% and only 9.6% had radiological evidence of pneumonia. In a study in Pakistani children with non-severe pneumonia, the radiological evidence of pneumonia was only found in 14% of cases as per WHO criteria [29]. A similar finding has been reported from a study done in Ethiopia which showed that 51.6% of the children diagnosed with severe pneumonia had no radiological evidence of pneumonia [30]. Studies from developed countries like the USA have also revealed similar findings in pediatric CAP [31,32]. Children with normal X-ray and low clinical suspicion of pneumonia can be observed without antibiotic therapy [33].

Our study had some strengths. First, we had a large sample size covering a period of three years. We used data collected routinely in our setting and hence the findings reflect the programmatic realities on the ground. Since the study site is a WHO surveillance center, the quality of the routine data related to pneumonia was good with periodic validation by dedicated and trained staff. Hence, the findings are valid and reliable.

We also had some limitations. First, since the study was conducted in a single tertiary care hospital, findings are not widely generalizable. Second, we might have overestimated the positive outcomes because of our strict exclusion criteria which excluded many children with more severe disease. Third, the information on additional supportive treatments provided (such as oxygen therapy, nebulization with beta-2 agonists, and others) and the main reason for admission were not captured in the electronic database. Hence, we are unable to report on these variables.

Despite these limitations, there are many implications for policy and practice. First, the majority of children with pneumonia admitted to a tertiary care hospital in a non-ICU setting can be treated effectively with ampicillin monotherapy without the need for an upgrade to second-and third-line antibiotics. The effectiveness of ampicillin remained high even in children with prehospital antibiotic use. These findings endorse AAP’s recommendation and its applicability to resource-limited settings like Nepal.

Second, the majority of the children did not have severe pneumonia and could have been managed using oral antibiotics at primary health centers without the need to visit a tertiary care center like Patan hospital. Therefore, there is a need to create awareness among the public to seek services from primary health centers. Strengthening the primary care will also help to decongest the tertiary care centers. Even in tertiary care settings, children with non-severe CAP may be treated with oral antibiotics.

Third, there were many antibiotic regimens used for treating children, which might be due to the unavailability of written protocol for the treatment of pediatric CAP. Hence, we recommend that a standard treatment protocol for CAP be developed and incorporated into the existing pediatric treatment protocol and implemented urgently. The protocol may also be displayed at several places in the hospital for easy reference. A follow-up study may be conducted to assess changes in practice.

Fourth, the yield of blood culture was extremely low and chest radiographs were normal in the majority of the children. Therefore, we need to reevaluate if these investigations should be a routine practice for all admitted cases of pneumonia, as these can increase the costs of treatment. Chest radiographs may be helpful in detecting the complications and may be considered in severe cases of pneumonia, rather than routinely in all children.

Finally, the cause of pneumonia could not be established in all the children due to the unavailability of point-of-care, rapid diagnostics. Given the presence of wheezing, low yield of blood culture, and normal chest radiography in the majority of children, we hypothesize that many of the pneumonia cases had viral etiology. This hypothesis is supported by the findings of a previous study from the same hospital [5]. Given the lack of diagnostic capacity in the hospital to confirm viral etiology of pneumonia, the treating pediatricians presumed bacterial etiology and treated every child with antibiotics. Availability of tests like polymerase chain reaction to detect pathogens from respiratory samples like sputum, nasopharyngeal secretions, or biomarkers from blood to differentiate the bacterial or viral etiology can help the pediatricians to decide whether to use antibiotics or not in children with clinically diagnosed pneumonia. This will help in the rational use of antibiotics (including a decision not to use antibiotics in children with viral pneumonia) and the prevention of antimicrobial resistance in the future.

## 5. Conclusions

In conclusion, we found that nearly seven in ten under-five children hospitalized with pneumonia in a tertiary care hospital in Nepal were managed with ampicillin monotherapy with good treatment outcomes. The outcomes remained good even in children with prehospital antibiotic use. Very few children needed second-and third-line antibiotics. This study highlights the effectiveness of ampicillin monotherapy in the treatment of community-acquired pneumonia in hospitalized children in a non-ICU setting.

## Figures and Tables

**Table 1 tropicalmed-06-00055-t001:** Sociodemographic and clinical characteristics of children (aged 2–59 months) with community-acquired pneumonia admitted to Patan hospital, Nepal during 2017–2019 (*n* = 659).

Characteristic	Number	Percentage
**Age (months)**		
2–11	306	(46.4)
12–23	155	(23.5)
24–35	108	(16.4)
36–47	61	(9.3)
48–59	29	(4.4)
**Gender**		
Female	376	(57.1)
Male	283	(42.9)
**Residence**		
Lalitpur	488	(74.1)
Bhaktapur	23	(3.5)
Kathmandu	80	(12.1)
Others	68	(10.3)
**History of PCV * vaccination**		
Received complete immunization as per age	447	(67.8)
Incomplete immunization as per age	74	(11.2)
Immunized but the number of doses not available	81	(12.3)
No immunization received	57	(8.6)
**History of Hib vaccination**		
Received complete immunization as per age	316	(48.0)
Incomplete immunization as per age	11	(1.7)
Immunized but the number of doses not available	319	(48.4)
No immunization received	13	(2.0)
**Nutritional status**		
Normal	477	(72.4)
Stunting	78	(11.8)
Wasting	60	(9.1)
Stunting and wasting	44	(6.7)
**WHO severity classification**		
No pneumonia	255	(38.7)
Pneumonia	231	(35.1)
Severe pneumonia	173	(26.3)
**Chest radiology**		
Normal	436	(66.2)
Infiltrates	139	(21.1)
End-point consolidation	74	(11.2)
Uninterpretable	10	(1.5)
**Prehospital antibiotic use**		
Yes	199	(30.2)
No	385	(58.4)
Unknown	75	(11.4)

* PCV—Pneumococcal conjugate vaccine; Hib—*Hemophilus Influenzae* B vaccine.

**Table 2 tropicalmed-06-00055-t002:** Patterns of antibiotic treatment received during hospitalization by children (aged 2–59 months) with community-acquired pneumonia admitted to Patan hospital, Nepal during 2017–2019.

Treatment Regimen	Prehospital Antibiotic Use (199)		No Prehospital Antibiotic Use (385)		Unknown Status (75)		Total (659)	
	*n*	(%)	*n*	(%)	*n*	(%)	*n*	(%)
AMP	127	(63.8)	282	(73.2)	49	(65.3)	458	(69.5)
AMP + AMK	13	(6.5)	16	(4.1)	4	(5.3)	33	(5.0)
AMP + AZT	4	(2.0)	11	(2.9)	1	(1.3)	16	(2.4)
AMP +CFTR	0	(0.0)	2	(0.5)	3	(4.0)	5	(0.8)
AMP + OFX	0	(0.0)	2	(0.5)	0	(0.0)	2	(0.3)
AMP + CLX	0	(0.0)	8	(2.1)	1	(1.3)	9	(1.4)
AMP + CFTR + CLP	1	(0.5)	0	(0.0)	0	(0.0)	1	(0.2)
AMP + AMK + CFTX	1	(0.5)	2	(0.5)	1	(1.3)	4	(0.6)
CFTR	36	(18.1)	37	(9.6)	8	(10.7)	81	(12.3)
CFTR + CLX	5	(2.5)	14	(3.6)	3	(4.0)	22	(3.3)
CFTR + AZT	2	(1.0)	2	(0.5)	1	(1.3)	5	(0.8)
CFTR + MTRZ	1	(0.5)	0	(0.0)	0	(0.0)	1	(0.2)
CFTR + CIP	1	(0.5)	0	(0.0)	0	(0.0)	1	(0.2)
CFTR + CLX + CLP	0	(0.0)	0	(0.0)	1	(1.3)	1	(0.2)
CLX	0	(0.0)	1	(0.3)	0	(0.0)	1	(0.2)
CLX + CLP	0	(0.0)	0	(0.0)	1	(1.3)	1	(0.2)
CLX + MTRZ	1	(0.5)	0	(0.0)	0	(0.0)	1	(0.2)
CLX + GEN	0	(0.0)	1	(0.3)	0	(0.0)	1	(0.2)
AZT	0	(0.0)	2	(0.5)	0	(0.0)	2	(0.3)
AZT + CIP	1	(0.5)	0	(0.0)	0	(0.0)	1	(0.2)
AZT + CLP + VAN	1	(0.5)	0	(0.0)	0	(0.0)	1	(0.2)
CLP	3	(1.5)	1	(0.3)	1	(1.3)	5	(0.8)
MERO	0	(0.0)	0	(0.0)	1	(1.3)	1	(0.2)
CIP	0	(0.0)	1	(0.3)	0	(0.0)	1	(0.2)
CFTX	1	(0.5)	0	(0.0)	0	(0.0)	1	(0.2)
Not recorded	1	(0.5)	3	(0.8)	0	(0.0)	4	(0.6)

AMP—Ampicillin; AMK—Amikacin; AZT—Azithromycin; CFTR—Ceftriaxone; OFX—Ofloxacin; CLX—Cloxacillin; CLP—Chloramphenicol; CFTX—Cefotaxime; MTRZ—Metronidazole; CIP—Ciprofloxacin; GEN—Gentamicin; VAN—Vancomycin; MERO—Meropenem.

**Table 3 tropicalmed-06-00055-t003:** Proportion of children who required switch to second-line and third-line antibiotics among children (aged 2–59 months) with community-acquired pneumonia admitted to Patan hospital, Nepal during 2017–2019.

Treatment Group (Based on First-Line Treatment)	Antibiotic Upgraded to			Total *n* (%)
	None *n* (%)	Second-Line *n* (%)	Third-Line *n* (%)	
Ampicillin alone	403 (88.0)	53 (11.6)	2 (0.4)	458 (100)
Ceftriaxone alone	77 (95.1)	4 (4.9)	0 (0.0)	81 (100)
Other regimens	100 (83.3)	15 (12.5)	5 (4.2)	120 (100)
Total	580 (88.0)	72 (10.9)	7 (1.1)	659 (100)

**Table 4 tropicalmed-06-00055-t004:** Overall usage of antibiotics among children (aged 2–59 months) with CAP admitted to Patan hospital, Nepal during 2017–2019 (*n* = 659).

Antibiotic	*n*	(%) *
Ampicillin	530	(80.4)
Ceftriaxone	157	(23.8)
Cloxacillin	44	(6.7)
Amikacin	38	(5.8)
Azithromycin	31	(4.7)
Cloramphenicol	26	(3.9)
Cefotaxim	7	(1.1)
Vancomycin	6	(0.9)
Ofloxacin	6	(0.9)
Meropenem	5	(0.8)
Linezolid	2	(0.3)
Others #	9	(1.4)

* Percentages are calculated using a total of 659 children as the denominator. # Cotrimoxazole, Clindamycin, Durataz (Piperacillin sodium and Tazobactam sodium), Ciprofloxacin, Gentamicin, Metronidazole.

## Data Availability

The data that support the findings of this study are available from the corresponding author, B.P., upon reasonable request.

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
