# Peer review of "Antibiotic Use and Treatment Outcomes among Children with Community-Acquired Pneumonia Admitted to a Tertiary Care Public Hospital in Nepal"

_tropicalmed, 2021, doi:10.3390/tropicalmed6020055_

Round 1
Reviewer 1 Report
I want to congratulate the authors for doing a retrospective review on antibiotics practices in childhood CAP in their tertiary hospital. As CAP is a major contributor to the < 5 y morbidity and mortality, adequate treatment will prevent more severe disease. In the era of growing AMR, reduction of AB use, and/or narrowing spectrum is essential.
My major concern is that every child admitted is treated with antibiotics, while etiology < 5 is mostly viral. Although authors reflect on this in the end of the discussion, this could be more emphasized throughout the manuscript. Do the outcomes remain good because majority of children in fact do not need antibiotics? For me this is not clear. I have some comments and questions for the authors to improve their paper.
- Introduction:
Is there any information on etiology of CAP in your region? And about pneumococcal resistance? At least, somewhere it should be stated that CAP < 5 years old is often viral in origin, especially in countries with Pneu and Hib vaccinations policies.
The objectives are clearly described.
- Study design: it should be added that it is a retrospective study.
In- and exclusion criteria are clearly described, the WHO definition of pneumonia is missing while authors refer to this definition in the results.
- Results
Line 166: how does it come that children with CAP do not meet pneumonia criteria?
Strikingly, there are many children with a normal CXR. What did these children have? Viral wheezing? URTI? Please elaborate in the discussion.
Table 1: how many children were requiring O2-therapy? Additional therapy with beta-agonist nebulization? Other therapies next to antibiotics? What was the main reason for admission?
It is expected that majority of blood cultures would be negative, since < 5 y pneumonia is often viral in origin. For me, it is unclear what part of children were admitted with bronchiolitis or viral wheezing. Authors should definitely elaborate on this in the discussion or table 1.
With respect to antibiotic switch, what were the criteria for switching? Did these children really have an indication for antibiotic use?
Table 2 is very large. The data might be summarized in antibiotic groups, e.g. penicillins (+additional); cephalosporins (+ additional).
Were all children treated with IV-antibiotics or also oral antibiotics? Clarify in the text and discuss: it is well described that in children who are not septic and able to have intake, oral antibiotics are equivalent to iv antibiotics even in a hospital setting. Especially now it is proven that in almost all children blood cultures are negative.
Table 4: what is the rationale behind using the rescue antibiotics? Is it local flavor, or was it discussed with a microbiologist/ID specialist? Sputum/local cultures?
4. Discussion
Authors nicely discuss the data and fortunately, majority of children has received ampicillin monotherapy with good clinical effect. They nicely compare findings with literature (e.g. CXR). They might elaborate on that pneumonia is a clinical diagnosis and CXR does not help to rule out pneumonia, as shown in the results. Only to show complications.
Authors should not only discuss ampicillin monotherapy is sufficient for most cases but also discuss overtreatment in their study group. In this hospitalized group of children, antibiotic therapy could probably by reduced or given orally in many cases. But more information on the study group is needed. Authors could put emphasis on CXR and blood culture won’t help in diagnosis of etiology but only in the most severe or complicated cases and refer to literature. Effort could better be put in reliable inflammatory markers and/or a rapid RSV antigen test to reduce antibiotic use.
Author Response
Reviewer 1
I want to congratulate the authors for doing a retrospective review on antibiotics practices in childhood CAP in their tertiary hospital. As CAP is a major contributor to the < 5 y morbidity and mortality, adequate treatment will prevent more severe disease. In the era of growing AMR, reduction of AB use, and/or narrowing spectrum is essential.
My major concern is that every child admitted is treated with antibiotics, while etiology < 5 is mostly viral. Although authors reflect on this in the end of the discussion, this could be more emphasized throughout the manuscript. Do the outcomes remain good because majority of children in fact do not need antibiotics? For me this is not clear. I have some comments and questions for the authors to improve their paper.
Response: Thanks for your appreciation and the comment about viral aetiology of pneumonia. We agree with this comment and have made changes throughout the paper reflecting this stance. Yes, it is very much possible that many of the children might not have needed antibiotics, but given the lack of diagnostic capacity in the hospital to confirm viral aetiology of pneumonia, the policy is to treat every child with antibiotics. We provide a point-by-point response to all the comments below.
- Introduction:
Is there any information on etiology of CAP in your region? And about pneumococcal resistance? At least, somewhere it should be stated that CAP < 5 years old is often viral in origin, especially in countries with Pneu and Hib vaccinations policies.
Response: Thanks for your query. Yes, there are previous studies from South-East Asian region which show that majority of the CAP in under-five children are caused by viruses, especially in areas with good access to HiB and pneumococcal vaccines. We have now included this statement in the introduction and cited the additional references.
The objectives are clearly described.
- Study design: it should be added that it is a retrospective study.
Response: Thank you. We have followed the STROBE guidelines in reporting this study. As per STROBE guidelines, the use of the term ‘retrospective’ is discouraged since this has been interpreted differently in the literature. Hence, we have deliberately not used the term in our paper. Instead, we clearly mention that our study is a hospital-based cross sectional study involving an analysis of routinely collected secondary data (which is more specific and unambiguous than the use of the term retrospective). Hence, we have not made any changes to the paper.
In- and exclusion criteria are clearly described, the WHO definition of pneumonia is missing while authors refer to this definition in the results.
Response: Thank you. Since our study population was chosen based on the IDSA definition of pneumonia and not the WHO definition, we have not mentioned it under study population. However, since classification of severity was done based on WHO definition, we have revised the methods section (subsection 2.4 data variables and sources) to explicitly mention that we have used the WHO definition. We have also added a reference in support of it.
- Results
Line 166: how does it come that children with CAP do not meet pneumonia criteria?
Response: Thank you. Since our study population was chosen based on the IDSA definition of pneumonia and not the WHO definition, about 40% of children did not fulfil the WHO criteria for defining pneumonia. As per WHO criteria, presence of tachypnoea is a mandatory criterion for diagnosing pneumonia, which is not the case as per IDSA guidelines. We hope this clarifies the apparent inconsistency.
Strikingly, there are many children with a normal CXR. What did these children have? Viral wheezing? URTI? Please elaborate in the discussion.
Response: Thank you. We want to clarify that all the children included in the study fulfilled the clinical criteria for pneumonia as per IDSA classification. It is possible some of the children had co-existing URTI, but none had URTI alone. Wheezing was one of the signs identified in 32% of children, though this does not mean that the child did not have pneumonia. This once again points towards a possible viral aetiology of pneumonia. We have added this point in the discussion.
Table 1: how many children were requiring O2-therapy? Additional therapy with beta-agonist nebulization? Other therapies next to antibiotics? What was the main reason for admission?
Response: Thank you for the comment. Unfortunately, we do not have information on additional treatments provided (like oxygen therapy, beta-agonist nebulization etc.) in our database. The main reason for admission was also not documented systematically in the patient files. Hence, we will not be able to report it in the paper. This is a limitation and we have now revised the limitations section of the paper to include this.
It is expected that majority of blood cultures would be negative, since < 5 y pneumonia is often viral in origin. For me, it is unclear what part of children were admitted with bronchiolitis or viral wheezing. Authors should definitely elaborate on this in the discussion or table 1.
Response: Thank you. Wheezing was one of the signs identified in 32% of children. This once again points towards a possible viral aetiology of pneumonia. We have added this into our results and discussion.
With respect to antibiotic switch, what were the criteria for switching? Did these children really have an indication for antibiotic use?
Response: Thank you. If there was clinical deterioration within 48 hours or clinical non-response even after 48 hours of first-line treatment, the treatment was switched to second-line treatment. If there was clinical deterioration within 48 hours or clinical non-response even after 48 hours of second-line treatment, the treatment was switched to third-line treatment. This has already been mentioned in the methods section (under 2.2.3). As mentioned above, given the lack of diagnostic capacity in the hospital to confirm viral aetiology of pneumonia, the treating paediatricians presumed bacterial aetiology and treated every child with antibiotics.
Table 2 is very large. The data might be summarized in antibiotic groups, e.g. penicillins (+additional); cephalosporins (+ additional).
Response: Thank you for the comment. We feel it is important to highlight the multiple regimens being used in the hospital and hence we want to retain the table in its current form. We feel improper formatting is one of the reasons why table 2 is looking very large. We have now modified Table 2 using acronyms for the drugs instead of the full names in the table and have expanded the acronyms in the footnotes of the table.
Were all children treated with IV-antibiotics or also oral antibiotics? Clarify in the text and discuss: it is well described that in children who are not septic and able to have intake, oral antibiotics are equivalent to iv antibiotics even in a hospital setting. Especially now it is proven that in almost all children blood cultures are negative.
Response: Thank you for the comment. All children were initiated on treatment using IV antibiotics except Azithromycin which was always given orally. Based on the response to treatment, the treatment was changed from IV to oral antibiotics in some cases. However, this was not systematically documented and hence we are unable to report the numbers. We agree that oral antibiotics are as effective as intravenous antibiotics in children with non-severe pneumonia. We have added this aspect in the discussion.
Table 4: what is the rationale behind using the rescue antibiotics? Is it local flavor, or was it discussed with a microbiologist/ID specialist? Sputum/local cultures?
Response: Thank you for the comment. The rationale for using reserve antibiotics was based on history of prehospital antibiotic use, severity of illness and clinical non-response to first-line treatment.
- Discussion
Authors nicely discuss the data and fortunately, majority of children has received ampicillin monotherapy with good clinical effect. They nicely compare findings with literature (e.g. CXR). They might elaborate on that pneumonia is a clinical diagnosis and CXR does not help to rule out pneumonia, as shown in the results. Only to show complications.
Response: Thank you for the comment. We have added a line in discussion to emphasise that CXR may not always be useful to diagnose pneumonia, but helpful to detect complications.
Authors should not only discuss ampicillin monotherapy is sufficient for most cases but also discuss overtreatment in their study group. In this hospitalized group of children, antibiotic therapy could probably by reduced or given orally in many cases. But more information on the study group is needed. Authors could put emphasis on CXR and blood culture won’t help in diagnosis of etiology but only in the most severe or complicated cases and refer to literature. Effort could better be put in reliable inflammatory markers and/or a rapid RSV antigen test to reduce antibiotic use.
Response: Thanks for your comments. We have now expanded our discussion to include possible overtreatment, potential use of oral antibiotics in non-severe cases and better diagnostics for viral pneumonia.
Reviewer 2 Report
Estimated Authors,
Estimated Editors,
I've read with great interest the paper from the group lead by Pokhrel, and supported by WHO, on the antibiotic use and treatment outcomes among children with community-acquired pneumonia admitted to a tertiary care public hospital in Nepal.
Despite the inherent limitations (mainly, being a study performed in settings substantially inconsistent with the medical resources usually available in Nepal), that have been frankly and properly addressed by Authors in the final section of the paper, I think that this paper may be of certain interest for the readers of TropMed, but some improvements are required.
My main concern is that this paper is a descriptive report on antimicrobial resistances in a very specific setting. It would be ever more interesting whether the AMR (even if regrouped in broader categories in order to be more easily handled) you identified were assessed in terms of respective risk factors (see Table 1). As the study includes a relatively large population, both univariate and multivariate analyses (e.g. AMR yes vs. no; shifting to 2nd / 3rd line drug vs. no) may bear significant data of certain interest for the readers of TropMed.
Another issue I suggest you to clarify is the definition of antimicrobial treatment of 1st, 2nd and 3rd line. More precisely, this definition is astonishing fluid, as depends on the settings, the economic requirements, and obviously the AMR even at the hospital / ward level. In your study, the first line antibiotic was defined as the drug used during the first 48 hrs. Please explain whether the choice of the antibiotic was associated or not with a preventive assessment other than the recommendation to follow WHO guidelines, and reformulate this section accordingly.
Author Response
Reviewer 2
I've read with great interest the paper from the group lead by Pokhrel, and supported by WHO, on the antibiotic use and treatment outcomes among children with community-acquired pneumonia admitted to a tertiary care public hospital in Nepal.
Despite the inherent limitations (mainly, being a study performed in settings substantially inconsistent with the medical resources usually available in Nepal), that have been frankly and properly addressed by Authors in the final section of the paper, I think that this paper may be of certain interest for the readers of TropMed, but some improvements are required.
Response: Thanks for your comments and appreciation.
My main concern is that this paper is a descriptive report on antimicrobial resistances in a very specific setting. It would be ever more interesting whether the AMR (even if regrouped in broader categories in order to be more easily handled) you identified were assessed in terms of respective risk factors (see Table 1). As the study includes a relatively large population, both univariate and multivariate analyses (e.g. AMR yes vs. no; shifting to 2nd / 3rd line drug vs. no) may bear significant data of certain interest for the readers of TropMed.
Response: Thanks for your comment. Unfortunately, the yield of blood culture was very low (only 5 patients were culture positive whom three were CoNS, considered as commensals and two were pathogens one Streptococcus pneumoniae, one Streptococcus viridans). Both streptococcus pneumoniae and streptococcus viridans were fully sensitive to pencillin group of drugs including ampicillin. Hence there was not even a single case of antimicrobial resistance. So, the suggested analysis cannot be carried out.
Another issue I suggest you to clarify is the definition of antimicrobial treatment of 1st, 2nd and 3rd line. More precisely, this definition is astonishing fluid, as depends on the settings, the economic requirements, and obviously the AMR even at the hospital / ward level. In your study, the first line antibiotic was defined as the drug used during the first 48 hrs. Please explain whether the choice of the antibiotic was associated or not with a preventive assessment other than the recommendation to follow WHO guidelines, and reformulate this section accordingly.
Response: Thanks for your comment. The choice of antibiotic was based on the clinical discretion of the treating paediatrician, history of pre-hospital antibiotic use and severity of illness in the child. We have added a line clarifying this in the methods.
Round 2
Reviewer 1 Report
Dear authors,
I am positively surprised by the quick reply and incorporated changes into the manuscript. My major concerns have been addressed and in my opinion the manuscript strongly improved by discussing the probable etiology of many of these CAP cases (viral), antibiotic policies (need for guideline and option for oral AB, even in a hospitalized setting) and discouraging the standard performance of BC and CXR's.
In my opinion, it is acceptable for publication now.
All the best with your antibiotic stewardship work!